# RLADAPTER: BRIDGING LARGE LANGUAGE MODELS TO REINFORCEMENT LEARNING IN OPEN WORLDS

## ABSTRACT

While reinforcement learning (RL) shows remarkable success in decision-making problems, it often requires a lot of interactions with the environment, and in sparse-reward environments, it is challenging to learn meaningful policies. Large Language Models (LLMs) can potentially provide valuable guidance to agents in learning policies, thereby enhancing the performance of RL algorithms in such environments. However, LLMs often encounter difficulties in understanding downstream tasks, which hinders their ability to optimally assist agents in these tasks. A common approach to mitigating this issue is to fine-tune the LLMs with task-related data, enabling them to offer useful guidance for RL agents. However, this approach encounters several difficulties, such as inaccessible model weights or the need for significant computational resources, making it impractical. In this work, we introduce RLAdapter[1], a framework that builds a better connection between RL algorithms and LLMs by incorporating an adapter model. Within the RLAdapter framework, fine-tuning a lightweight language model with information generated during the training process of RL agents significantly aids LLMs in adapting to downstream tasks, thereby providing better guidance for RL agents. We conducted experiments to evaluate RLAdapter in the Crafter environment, and the results show that RLAdapter surpasses the SOTA baselines. Furthermore, agents under our framework exhibit common-sense behaviors that are absent in baseline models.

## 1 INTRODUCTION

Reinforcement learning (RL) has demonstrated impressive capabilities in decision-making problems (Kaelbling et al., 1996; Sutton & Barto, 2018). The strength of RL algorithms is most evident when agents consistently receive clear and regular rewards that guide them toward the targeted behaviors (Ladosz et al., 2022; Eschmann, 2021). However, designing these reward functions is far from straightforward. It often requires meticulous engineering and access to a comprehensive set of information. This challenge becomes even more pronounced in sparse-reward environments.

To address these challenges, there is a growing interest in intrinsically motivated RL methods (Aubret et al., 2019). These methods augment the reward with additional objectives, often drawing inspiration from novelty, uncertainty, surprise, or predictive deviations. However, another problem arises: not all novelties or uncertainties necessarily align with the agent's goal or have intrinsic value (Burda et al., 2019; Ladosz et al., 2022). In this context, Large Language Models (LLMs) (Wei et al., 2022) present a promising direction. These models, pre-trained on massive corpus data, encapsulate a vast repository of human knowledge. While the use of LLMs to guide RL agents (Du et al., 2023) sounds promising, there are also challenges. Despite their extensive knowledge, LLMs often face difficulties in understanding specific downstream tasks (Bommasani et al., 2021). This limitation potentially undermines their efficacy in assisting RL agents seamlessly.

Some studies have explored the use of task-related data to fine-tune LLMs, aiming to better assist RL algorithms in specific tasks (Nottingham et al., 2023). However, such approaches often encounter practical challenges, such as inaccessible LLM weights or intensive computational demands. More-over, fine-tuning LLMs may lead to decreases in their generalization performance, making their

---

[1]The code for RLAdapter is anonymously released at: https://anonymous.4open.science/r/ICLR2024_RLAdapter

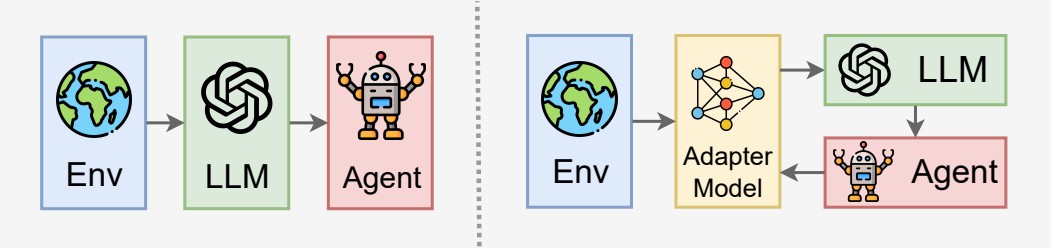

**Figure 1.** The core difference between our method (*right*) and traditional LLM-assisted RL decision-making methods (*left*). The key distinction primarily lies in the integration of the adapter model. This model enhances the feedback between RL agents and LLMs, effectively bridging the gap between LLMs and downstream tasks.

deployment across diverse environments challenging. In light of these issues, we do not focus on directly modifying LLMs, but instead consider adding adjustable modules to help LLMs adapt to the environment. This insight motivates us to propose the **RLAdapter** framework, designed to enhance collaboration between RL algorithms and LLMs. As illustrated in Figure 1, the key characteristic that distinguishes RLAdapter from existing methods (Choi et al., 2022; Du et al., 2023) is the integration of a lightweight adapter model. Enriched with feedback and information from the RL training phase, this adapter model adeptly prompts LLMs like GPT-4, enabling a refined understanding of tasks and agents' learning capabilities without modifying the LLMs' weights. In this way, RLAdapter effectively bridges the gap, establishing a more effective connection between RL algorithms and LLMs, thereby addressing the aforementioned challenges.

We empirically evaluate RLAdapter in the *Crafter* environment (Hafner, 2021) across 22 different tasks. Results highlight RLAdapter's superior performance in comparison to state-of-the-art baselines, while also showcasing agent behaviors grounded in common sense. Our main contributions are summarized as follows:

- We propose a novel framework to align LLMs to downstream tasks and to guide RL agents to effectively learn difficult tasks.

- We design the adapter model that correlates its own update with the learning progress of the agent and correspondingly generates appropriate prompts for LLMs, thereby forming a feedback loop together with LLMs and RL agents.

- We rigorously evaluate our framework's efficacy in the open-world game *Crafter* and provide a comprehensive analysis of the experimental results and an interpretation of the agent behavior grounded in common sense

## 2 RELATED WORK

**Large Language Models (LLMs).** Recent advances in natural language processing have led to an era dominated by large language models (LLMs). Central to this advancement is the GPT family, renowned for its impressive versatility across tasks. Alongside GPT, there are other prominent LLMs that have also made significant impacts (Chowdhery et al., 2022; Thoppilan et al., 2022). A turning point in the evolution of LLM is the adoption of instruction tuning (Ouyang et al., 2022). When LLMs are trained with human instructions, there is a remarkable improvement in their adaptability, particularly in challenging scenarios such as zero-shot and few-shot learning.

With the open source of some LLMs (Zeng et al., 2022; Touvron et al., 2023a), researchers have started to attempt fine-tuning them using data from downstream tasks (Wu et al., 2023a). This can lead to significant performance improvements in the corresponding tasks, but it also results in a severe decrease in the generalization performance of LLMs (Wang et al., 2022). In our work, although we also fine-tune the adapter model, we do not restrict it to specific datasets. Instead, we dynamically fine-tune it through RL agents' real-time feedback. Additionally, the selection of a lightweight model ensures that our approach can be easily transferred to new environments.

**LLMs for RL.** Using language to represent goals allows for the harnessing of large language models trained on expansive corpora. The use of LM-encoded goal descriptions has been shown to improve the generalization of instruction-following agents (Chan et al., 2019; Hill et al., 2020). The vast knowledge encapsulated in pretrained LLMs provides nuanced guidance via sub-goals and sub-policies (Lynch & Sermanet, 2020; Sharma et al., 2021). Subsequent studies have tried to link these sub-policies to tackle more intricate tasks (Huang et al., 2022a;b). Moreover, several methods use LLMs to offer intrinsic rewards, increasing the efficiency of RL learning (Choi et al., 2022; Du et al., 2023). However, despite the good performance of these methods in certain simple environments like text-based games, they often encounter scalability and generalization challenges in sophisticated environments (Zhong et al., 2021; Wang & Narasimhan, 2021). In our work, we seek to make LLM more flexible and convenient to provide useful assistance to RL algorithms in complex environments.

**LLMs for Open-World Games.** Open-world games present more challenges, such as managing long horizons (Hafner, 2021) and prioritizing parallel objectives (Wang et al., 2023b). While some researchers employ LLMs for planning and guiding RL in such contexts (Du et al., 2023; Yuan et al., 2023; Tsai et al., 2023), they often rely on human-generated trajectories as context. The limited scope of these trajectories might restrict performance in unseen scenarios, often making them less effective than recent RL algorithms (Hafner et al., 2023), which do not utilize LLMs. In addition, there are also some methods that only use LLMs for decision-making (Wu et al., 2023b; Wang et al., 2023a). However, these methods tend to have intricate designs tightly coupled with specific environments and datasets, which can make them less transferable to different tasks. On the contrary, our method is free from such complexity. Its straightforward design ensures adaptability across diverse environments.

## 3 METHODOLOGY

### 3.1 PROBLEM FORMULATION

We consider a partially observable Markov decision process (POMDP), defined by the tuple $(\mathcal{S}, \mathcal{A}, \mathcal{P}, \Omega, \mathcal{O}, R, \gamma)$, where $s \in \mathcal{S}$ and $a \in \mathcal{A}$ represent the states and actions, respectively. $\mathcal{P}(s'|s, a)$ denotes the dynamics of the environment. The observation, represented as $o \in \Omega$, is derived via $\mathcal{O}(o|s, a)$. $R$ stands for the reward function, and $\gamma$ denotes the discount factor.

In such an environment, our objective is to train a policy, denoted as $\pi(a|o, g)$, to maximize the cumulative reward. Here, $g$ represents the sub-goals provided by Large Language Models (LLMs).

### 3.2 OVERALL FRAMEWORK

Pre-trained LLMs contain massive information and exhibit impressive zero-shot language understanding across various tasks. This ability can assist agents in rapidly grasping their current situation in complex environments, consequently preventing them from getting stuck in exploration dilemmas. By prompting the LLMs, we can get text-format sub-goals. These textual insights, once embedded, are concatenated with the observation and provided for the policy $\pi(a|o, g)$. Despite the powerful generalization capabilities of LLMs, their understanding of specific tasks is not always comprehensive. Directly using guidance generated from LLMs may not often result in a coherent grounding understanding in the observations. A typical solution involves fine-tuning the LLM using task-specific data. However, this solution can be computationally intensive. Moreover, black-box models, like GPT-4, may be challenging to fine-tune since accessing their weights is not feasible.

Given the challenges, we focus more on adding adjustable modules to assist LLMs in agilely adapting to the environment, rather than modifying the LLMs directly. A key insight is that even a lightweight language model, with the right fine-tuning, can excel at particular tasks (Zhang et al., 2023; Li et al., 2023). This motivates us to propose RLAdapter, as illustrated in Figure 2. The core component of RLAdapter is a lightweight language model that serves as an adapter. Instead of directly relaying information from the environment to the LLM, our method utilizes the adapter model to first analyze the agent's status and learning abilities. The adapter model then generates concise summaries and suggestions for the LLM, which in turn produces the final sub-goals. This structure allows the lightweight adapter model to continuously refine itself according to agent's feedback. This ensures that it can accurately summarize important information from both the agent and the

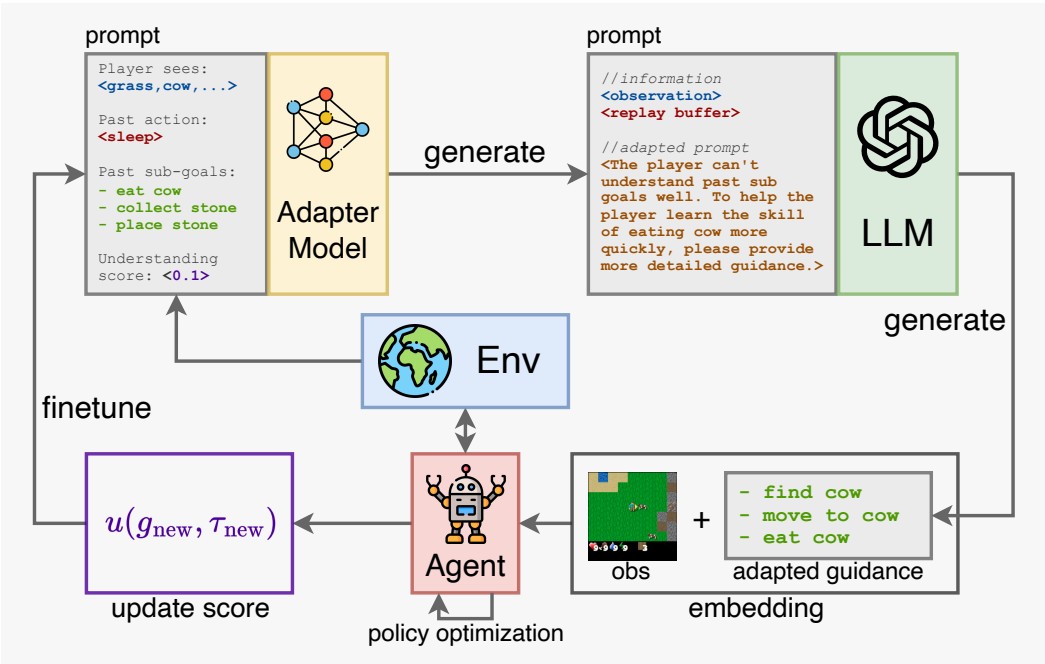

**Figure 2.** Overall framework of RLAdapter. In addition to receiving inputs from the environment and historical information, the prompt of the adapter model incorporates an understanding score. This score computes the semantic similarity between the agent's recent actions and the sub-goals suggested by the LLM, determining whether the agent currently comprehends the LLM's guidance accurately. Through the agent's feedback and continuously fine-tuning the adapter model, we can keep the LLM always remains attuned to the actual circumstances of the task. This, in turn, ensures that the provided guidance is the most appropriate for the agents' prioritized learning.

environment. Using its powerful summarization capabilities, the adapter efficiently communicates critical details to larger LLMs, enabling them to provide appropriate guidance to agents.

### 3.3 ADAPTER MODEL

The input information for the adapter model primarily comprises two aspects: the basic information about the environment and the agent's present understanding level of the language guidance. The basic information about the environment can be provided by the game engine or visual descriptors (Radford et al., 2021). Meanwhile, the agent's comprehension of the language guidance is characterized by computing the cosine similarity between the sub-goals provided by the LLM and the episode trajectory after they've both embedded:

$$u(g, \tau) \doteq \cos(f_{\text{emb}}(g) \cdot f_{\text{emb}}(\tau)) = \frac{f_{\text{emb}}(g) \cdot f_{\text{emb}}(\tau)}{\|f_{\text{emb}}(g)\| \cdot \|f_{\text{emb}}(\tau)\|}. \tag{1}$$

Here, $f_{\text{emb}}$ represents the encoder that embeds text information into vectors. In our implementation, we choose SentenceBert (Reimers & Gurevych, 2019) for embedding. A higher value of $u$ indicates that the agent's current actions align more closely with the language guidance, suggesting that the agent has a better understanding of the LLM. We then integrate both the understanding score $u$ and the environmental information from replay buffer $\mathcal{B}$ into the prompt for the adapter model, *i.e.*, `prompt`$(\mathcal{B}, u)$, where `prompt`$(\cdot)$ is the prompt template. When receiving `prompt`$(\mathcal{B}, u)$, the adapter model extracts the important details and, based on $u$, analyzes the agent's understanding capability. Subsequently, the adapter model generates summarized information $c \sim \mathcal{M}_{\text{ada}}(\texttt{prompt}(\mathcal{B}, u))$, where $\mathcal{M}_{\text{ada}}$ represent the adapter model. Similarly, the generated $c$ is passed through the template along with the information in the replay buffer as a prompt for the LLM, *i.e.*, `prompt`$(\mathcal{B}, c)$. In our implementation, the details of the prompt template can be referred to the example provided in Appendix B.

---

**Algorithm 1** Pseudo Code for RLAdapter

---

1: **Init:** Policy $\pi$; Buffer $\mathcal{B}$; Supervised fine-tuning (SFT) buffer $\mathcal{D}$; LLM generation interval $N_{\text{gen}}$; SFT interval $N_{\text{sft}}$; ROGUE-L threshold $\theta$.
2: $o_0 \leftarrow \texttt{env.reset()}, u_0 \leftarrow 0$
3: **for** $t = 0, 1, \dots$ **do**
4:     // generate with adapter model and LLM (with interval $N_{\text{gen}}$)
5:     **if** $t \,\% \, N_{\text{gen}} = 0$ **then**
6:         $c_t \leftarrow \mathcal{M}_{\text{ada}}(\texttt{prompt}(\mathcal{B}_t, u_t)), g_t \leftarrow \mathcal{M}_{\text{LLM}}(\texttt{prompt}(\mathcal{B}_t, c_t))$
7:     **else**
8:         $c_t \leftarrow c_{t-1}, \; g_t \leftarrow g_{t-1}$
9:     **end if**
10:    // interact with the environment
11:    $a_t \sim \pi(a_t | o_t, f_{\text{emb}}(g_t)), o_{t+1} \leftarrow \texttt{env.step}(a_t)$
12:    // update buffer and policy
13:    $\mathcal{B}_{t+1} \leftarrow \mathcal{B}_t \cup (o_t, a_t, o_{t+1}, r_t, g_t)$
14:    $\pi_{t+1} \leftarrow \texttt{update}(\pi_t, \mathcal{B}_{t+1})$
15:    // update understanding score and SFT buffer
16:    $u_{t+1} \leftarrow \cos(f_{\text{emb}}(g_t), f_{\text{emb}}(\tau)), \; \tau \sim \mathcal{B}_{t+1}$
17:    **if** $\texttt{ROGUE-L}(\mathcal{D}; [\texttt{prompt}(\mathcal{B}_t, u_{t+1}), c_t]) < \theta$ **then**
18:       $\mathcal{D} \leftarrow \mathcal{D} \cup [\texttt{prompt}(\mathcal{B}_t, u_{t+1}), c_t]$
19:    **end if**
20:    // SFT adapter model (with interval $N_{\text{sft}}$)
21:    **if** $t \,\% \, N_{\text{sft}} = 0$ **then**
22:       $\texttt{SFT}(\mathcal{M}_{\text{ada}}; \mathcal{D})$
23:    **end if**
24: **end for**

---

## 3.4 TRAINING PROCEDURE

During training, RL agents continuously explore, collect data, and improve their capabilities. This aligns with the adapter model's objective of continuously refining its understanding of the environment and the agent. This alignment can be ingeniously designed into a unified training procedure. Hence, we integrate the fine-tuning of the adapter model with the updates of the RL agents.

In detail, when receiving the adapted prompt $c \sim \mathcal{M}_{\text{ada}}(\texttt{prompt}(\mathcal{B}, u))$, the LLM generates $g \sim \mathcal{M}_{\text{LLM}}(\texttt{prompt}(\mathcal{B}, c))$, which is subsequently provided to the policy $\pi(a|o, g_{\text{emb}})$ for training, where $g_{\text{emb}}$ is the text embedding encoded by $f_{\text{emb}}$. Once updated, new trajectories yield a refreshed understanding score, $u_{\text{new}} \doteq u(g_{\text{new}}, \tau_{\text{new}})$. Then we construct a pair of linguistic data $l = [\texttt{prompt}(\mathcal{B}, u_{\text{new}}), \texttt{prompt}(\mathcal{B}, c)]$, for supervised fine-tuning (Gunel et al., 2020). In such a training process, $u_{\text{new}}$ plays a crucial role. Adding $u_{\text{new}}$ to the fine-tuning data enables the adapter model to refine its self-awareness of the effect on the generated $c$. Furthermore, to maintain diversity in the fine-tuning data pool, additional steps are taken for data filtering and post-processing. Specifically, the ROGUE-L similarity (Lin, 2004) is calculated between the new data and each entry in the data pool to decide its inclusion. The calculation details of ROGUE-L similarity can be found in Appendix C.

To ensure that the incorporation of the adapter model and LLM does not excessively burden the RL training in terms of time, and considering that completing each subgoal in an open-world game environment requires a sequence of consecutive steps, we do not query the language models at every step. Instead, we set a predetermined interval between queries, maintaining consistent guidance during the intervals. Similarly, the fine-tuning process also takes place at specified intervals. In line with our claim that only a lightweight adapter model is necessary, we utilized the 4-bit quantized version of the LLaMA2-7B model (Touvron et al., 2023b) as the base model and employed Qlora (Dettmers et al., 2023) for efficient fine-tuning.

For policy learning, we use the classic PPO algorithm (Schulman et al., 2017) as implementation. It is worth noting that RLAdapter can be flexibly combined with various RL algorithms and is not limited to PPO. The overall process of the algorithm can refer to Algorithm 1. Detailed parameters and settings can be found in Appendix A.

## 4 EXPERIMENT

Our experiments primarily aim to validate the following claims:

- The integration of the adapter model can enhance the Large Language Model's comprehension of downstream tasks and the agent's understanding capability, resulting in more meaningful guidance.
- Agents trained under the RLAdapter framework can exhibit superior performance and demonstrate behavior with more common sense.

### 4.1 EXPERIMENT SETTINGS

**Environment.** We conducted our experiments in the *Crafter* environment (Hafner, 2021). *Crafter* serves as a widely used benchmark for evaluating the decision-making capabilities of agents in open-world games. It comprises a $64 \times 64$ grid map, with each cell containing a predefined object (*e.g.* grass, water, wood) or an entity (*e.g.* player, zombie, skeleton). Agents can observe a local $9 \times 7$ area within this environment. In *Crafter*, there is no single main task for the agent to accomplish. Instead, the agent is expected to learn various skills to complete 22 different achievements across different levels. Moreover, the health and resources of the agent decrease over time, and they also face challenges like monster attacks. In such a setting, the agent not only needs to address the challenge of sparse rewards, but also consider how to survive as much as possible.

**Evaluation Metrics.** In the *Crafter* environment, every time an agent unlocks a new achievement, it receives a $+1$ reward. Additionally, when the agent gains or loses 1 health point, it is given a $+0.1/-0.1$ reward, respectively. The game also provides an overall score metric (Hafner, 2021) based on the success rate ($s_i \in [0, 100]$) of each achievement:

$$S \doteq \exp\left(\frac{1}{N}\sum_{i=1}^{N}\ln\left(1+s_i\right)\right) - 1, \tag{2}$$

where $N = 22$ denotes the number of achievements. We evaluated the performance of the methods using both reward and score metrics.

**LLM Choices.** In order to demonstrate the effective collaboration between RLAdapter and LLMs at different levels, we chose OpenAI's GPT-4 and GPT-3.5 as the LLM in RLAdapter for the experiment. We will also analyze the specific impacts caused by different LLM choices in the ablation study.

**Prompt Design.** For the adapter model, we extract information about objects in the observation and the agent's status from the game engine. Combined with the previously mentioned understanding score $u$, this serves as the prompt for the adapter model. The format is as follows: "`Player sees: <observations>; Past action: <past actions>; Past sub-goals: <last suggested sub-goals>; Understanding score: . Analyze the environment and the player's understanding capability, then generate concise summaries and suggestions about this player.`" For the LLM, the prompt is constructed based on the output of the adapter model: "`<output of the adapter model>. Based on the provided information, suggest 3 sub-goals that the player should accomplish next.`"

### 4.2 BASELINES

To demonstrate the superiority of RLAdapter, we compare it with the following methods:

- LLM-assisted solutions: SPRING (Wu et al., 2023b), ELLM (Du et al., 2023)
- Model-based RL method: Dreamer-V3 (Hafner et al., 2023)
- Classic RL algorithms: PPO (Schulman et al., 2017), Rainbow (Hessel et al., 2018)
- Intrinscial methods: RND (Burda et al., 2019), Plan2Explore (Sekar et al., 2020)

We also add human experts (Hafner, 2021), standalone GPT-4 (step-by-step instructions), and random policy as additional references.

**Table 1.** Performance comparison between RLAdapter and baselines in terms of score and reward metrics. The results of RLAdapter are obtained from 5 independent training trials. To match the settings of different baselines, we separately present the performance of RLAdapter with 1 million and 5 million training steps. The results indicate that the RLAdapter with GPT-4 outperforms all baselines, while the RLAdapter with GPT-3.5 is also close to SPRING, which is the SOTA method using GPT-4. Note that $\pm$ captures standard deviations.

| Method | Score | Reward | Training Steps |
|---|---|---|---|
| RLAdapter (w/ GPT-4) | $\mathbf{28.0} \pm 2.5\%$ | $\mathbf{12.8} \pm 1.9$ | $5 \times 10^6$ |
| SPRING (w/ GPT-4) | $27.3 \pm 1.2\%$ | $12.3 \pm 0.7$ | — |
| RLAdapter (w/ GPT-3.5) | $25.6 \pm 2.7\%$ | $12.4 \pm 1.8$ | $5 \times 10^6$ |
| RLAdapter (w/ GPT-4) | $15.4 \pm 2.2\%$ | $12.1 \pm 1.5$ | $1 \times 10^6$ |
| RLAdapter (w/ GPT-3.5) | $14.7 \pm 2.3\%$ | $11.8 \pm 2.1$ | $1 \times 10^6$ |
| DreamerV3 | $14.5 \pm 1.6\%$ | $11.7 \pm 1.9$ | $1 \times 10^6$ |
| ELLM | — | $6.0 \pm 0.4$ | $5 \times 10^6$ |
| PPO | $4.6 \pm 0.3\%$ | $4.2 \pm 1.2$ | $1 \times 10^6$ |
| Rainbow | $4.3 \pm 0.2\%$ | $5.0 \pm 1.3$ | $1 \times 10^6$ |
| Plan2Explore | $2.1 \pm 0.1\%$ | $2.1 \pm 1.5$ | $1 \times 10^6$ |
| RND | $2.0 \pm 0.1\%$ | $0.7 \pm 1.3$ | $1 \times 10^6$ |
| Human Experts | $50.5 \pm 6.8\%$ | $14.3 \pm 2.3$ | — |
| GPT-4 | $3.4 \pm 1.5\%$ | $2.5 \pm 1.6$ | — |
| Random | $1.6 \pm 0.0\%$ | $2.1 \pm 1.3$ | $1 \times 10^6$ |

## 4.3 RESULTS AND ANALYSIS

We compare the performance of various algorithms, including some methods that do not have open-source code. For these methods, we directly reference the results from their papers for comparison. To ensure fairness in the comparison, we set up two versions of RLAdapter with 1 million and 5 million training steps respectively to match the settings of these different baselines. The performance results are shown in Table 1.

The results show that when the training steps reach 1 million, RLAdapter with GPT-3.5 is sufficient to outperform the baselines with the same number of steps and ELLM with 5 million steps. When the number of steps reaches 5 million, RLAdapter with GPT-4 demonstrates better performance than all baselines, and RLAdapter with GPT-3.5 also matches SPRING in terms of reward metrics. Note that in the performance of RLAdapter and SPRING, $\pm$ captures different standard deviations. The performance variation of RLAdapter comes from 5 training trials, while that of SPRING comes from test trials. Although RLAdapter and SPRING are two conceptually different methods, making it difficult to choose the steps for the most appropriate comparison, the results are enough to demonstrate that RLAdapter can bring further performance improvements for LLMs like GPT-4. *Moreover, RLAdapter does not have expert-level data dependency like SPRING, which is another major advantage of RLAdapter.*

We also investigate the success rates of RLAdapter (with GPT-4) and two top-performing baselines (SPRING and DreamerV3) on 22 specific achievements. As shown in Figure 3, the results indicate that RLAdapter exhibits higher success rates in most achievements, except for "Defeat Skeleton" and "Defeat Zombie" where success rates are relatively lower. Through replay analysis, we find that this phenomenon is due to the fact that the policy trained by RLAdapter tends to believe that fighting monsters is not conducive to survival, so it prefers to avoid combat with them. This demonstrates that the collaboration between the adapter model and GPT-4 can bring about some unexpected but useful strategies. In addition, both RLAdapter and DreamerV3 have a certain success rate in "Place Stone", while SPRING has not learned at all. This clearly indicates that the lack of training data in the pure LLM method leads to its deficiency in specific tasks.

However, at the most difficult level of "Make Iron Pickaxe", "Make Iron Sword" and "Collect Diamond", all methods have a success rate of 0. This indicates that in future research, we still need to further improve current methods for such complex tasks.

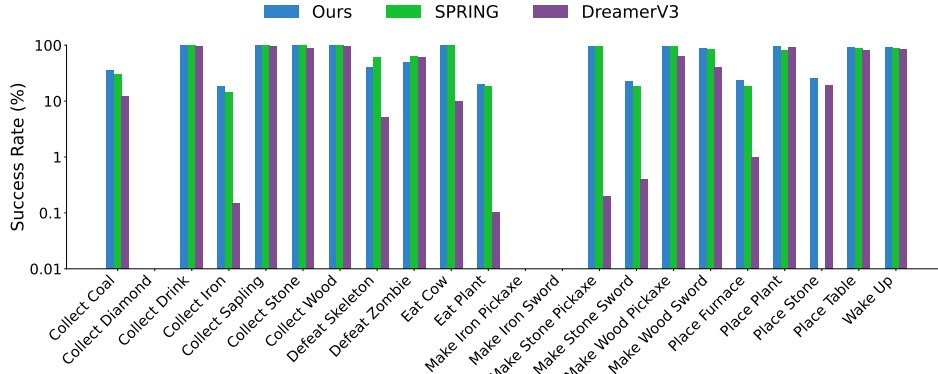

**Figure 3.** Log scale success rates of unlocking 22 different achievements. RLAdapter outperforms baselines in most achievements, with relatively lower success rates only in "Defeat Skeleton" and "Defeat Zombie". This is because the policy trained by RLAdapter tends to avoid combat.

**Table 2.** Ablation study results of RLAdapter. Achievement depth represents the highest level of achievement that agents can accomplish, with a maximum value of 7. These methods indicate the impact on performance caused by using different LLMs and whether or not to include the adapter model. All results are obtained from 5 independent trials.

| Method | Score | Reward | Achievement Depth |
|---|---|---|---|
| RLAdapter (GPT-4) | $28.0 \pm 2.5\%$ | $12.8 \pm 1.9$ | 6 |
| RLAdapter (GPT-3.5) | $25.6 \pm 2.7\%$ | $12.4 \pm 1.8$ | 6 |
| RLAdapter (GPT-4) w/o adapter model | $9.6 \pm 1.7\%$ | $8.7 \pm 1.4$ | 5 |
| RLAdapter (GPT-3.5) w/o adapter model | $7.7 \pm 1.6\%$ | $6.4 \pm 1.5$ | 4 |
| GPT-4 | $3.4 \pm 1.5\%$ | $2.5 \pm 1.6$ | 1 |
| GPT-3.5 | $2.9 \pm 1.7\%$ | $2.7 \pm 1.9$ | 1 |

## 4.4 ABLATION STUDY

To investigate the contribution of each component within the RLAdapter framework, we conducted a series of ablation studies. Specifically, we compare the performances of using different LLMs in RLAdapter and removing the adapter model. In addition, we also use standalone GPT-4/GPT-3.5 as references. To ensure fair comparisons between them, we try to maintain consistency in their prompts as much as possible. Please refer to Appendix B for full prompts.

The results of these studies are shown in Table 2. From the results, we can observe that substituting GPT-4 with GPT-3.5 results in only minor performance degradation. However, when the adapter model is removed from RLAdapter (*i.e.*, degenerates into the typical method of directly assisting RL with LLMs), there are significant performance decreases. Interestingly, without the adapter model, GPT-4 outperforms GPT-3.5 in terms of achievement depth. However, with the inclusion of the adapter model, GPT-3.5 reaches the same achievement depth as GPT-4. This suggests that the adapter model plays a crucial role in leveraging the capabilities of LLMs. And according to the results of standalone GPT-4 and GPT-3.5, it is obvious that simply following LLM instructions is not sufficient for effective learning, underscoring the importance of combining LLMs and RL algorithms in intricate environments like open-world games.

## 4.5 AGENT BEHAVIORS GROUNDED IN COMMON SENSE

As discussed in Section 4.3, the policy trained by RLAdapter exhibits behaviors like avoiding combat. Although this may result in a partial performance decrease for the achievements "Defeat Skeleton" and "Defeat Zombie", it could be more advantageous for survival and better completion of other tasks. In this sense, RLAdapter demonstrates behaviors that align with human common sense. We

**Table 3.** Case study on agent behaviors grounded in common sense. These behaviors demonstrate the ability of the adapter model in uncovering human knowledge behind LLMs. For more specific details and descriptions about these cases, please refer to Appendix D.

| Case | Description | Explanation |
|---|---|---|
|  | The agent tends to place stones between itself and monsters to avoid combat at night (the number of monsters will increase). | Frequent combats are not conducive to maintaining health and can delay other tasks such as resource collection. Therefore, the agent chooses to avoid combat at the appropriate time. |
|  | The agent does not immediately place a workbench to craft tools and unlock achievements when it has abundant resources, but instead places the workbench when moving to resource-rich areas. | Placing the workbench in resource-rich areas can reduce the distance between collecting resources and crafting items, thus improving efficiency. |

further analyze additional replays and find other cases of human-like behavior in the policy trained by RLAdapter, which were not reported in previous work, as shown in Table 3. For more details about these behaviors, please refer to Appendix D.

In the two cases, RLAdapter demonstrates behaviors such as using stones to block monsters and extend survival time, as well as placing workbenches in resource-rich areas for more efficient resource utilization. These behaviors are not observed or reported in other baselines or in the version of RLAdapter w/o adapter model. This further demonstrates that the adapter model can better capture the agent's learning ability and uncover common-sense knowledge behind LLMs, prompting them to provide more useful and reasonable guidance for better decision-making.

## 5 LIMITATIONS AND FUTURE WORK

The main limitation of RLAdapter is that it still requires a certain level of pre-trained knowledge of the adapter model. If a small language model is used as an adapter, its language understanding ability may not be sufficient to provide the necessary analysis for the environment and agents. Additionally, although RLAdapter significantly improves the performance of RL algorithms, as analyzed in Section 4.3, there are still limitations when dealing with more complex tasks.

Nevertheless, the uncovering of knowledge about GPT-4 and other LLMs by the adapter model demonstrates promising prospects for filling the gap in LLMs' performances across various tasks. In future work, we will continue to explore this characteristic of the adapter model while also attempting to integrate LLM with RL algorithms more closely to address these limitations in complex environments.

## 6 CONCLUSIONS

In this work, we design the RLAdapter framework to facilitate better collaboration between LLMs and RL algorithms. The core of this method is the integration of an adapter model between RL algorithms and LLMs, which can enhance the feedback between agents and LLMs. We evaluated RLAdapter on the open-world game *crafter*. Experimental results demonstrate that RLAdapter not only improves the performance of RL algorithms, surpassing SOTA baselines, but also enables learned policies to exhibit behavior more aligned with common sense.

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

## A    IMPLEMENTATION DETAILS

### A.1    RL ALGORITHM

We use the classic PPO algorithm for policy learning in RLAdapter, and the hyperparameters are shown in Table 4. It is worth noting that RLAdapter can be flexibly combined with various RL algorithms and is not limited to PPO.

**Table 4.** Hyperparameters for PPO.

| Hyperparameter | Value |
|---|---|
| policy learning rate | 7e-4 |
| update epoch | 16 |
| $\gamma$ | 0.97 |
| $\varepsilon$ | 1e-8 |
| clip ratio | 0.1 |
| optimizer | Adam |

### A.2    ADAPTER MODEL

We use open-source LLaMA2-7B weight as initial weight for the adapter model. In order to reduce computational resources and time consumption, we perform 4-bit quantization on it. The SFT parameters of the adapter model are shown in Table 5.

**Table 5.** Hyperparameters for Supervised Fine-Tuning.

| Hyperparameter | Value |
|---|---|
| quant type | nf4 |
| learning rate | 2e-4 |
| batch size | 4 |
| gradient accumulation step | 1 |
| weight decay | 1e-3 |
| max grad norm | 0.3 |
| warmup ratio | 0.3 |
| lora alpha | 16 |
| lora dropout | 0.1 |
| lora r | 64 |
| $N_{\text{gen}}$ (w/ GPT3.5) | 10 |
| $N_{\text{gen}}$ (w/ GPT4) | 30 |
| $N_{\text{sft}}$ | 1e3 |

### A.3    OTHER SETTINGS

We call the API interfaces of OpenAI's gpt-4 and gpt-3.5-turbo models. The API parameters used are shown in Table 6.

**Table 6.** Hyperparameters for LLM.

| Hyperparameter | Value |
|---|---|
| temperature | 0.5 |
| top_p | 1.0 |
| max_tokens | 100 |

For text embedding, we choose the open-source paraphrase-MiniLM-L6-v2 as the encoder.

## B  FULL PROMPT DETAILS

In the following, we provide detailed prompts as well as corresponding example outputs for different methods.

### B.1  PROMPTS FOR ADAPTER MODEL IN RLADAPTER

System message:

```
You are a professional game analyst. A player is playing a game
similar to Minecraft. Available actions are:
<move_left, move_right, move_up, move_down, do, sleep, place_stone
, place_table, place_furnace, place_plant, make_wood_pickaxe,
make_stone_pickaxe, make_iron_pickaxe, make_wood_sword,
make_stone_sword, make_iron_sword>.

You will get the player's observation, status information, and its
 understanding score of language guidance (between 0 and 1). You
are collaborating with another analyst, and you will be asked to
provide concise summaries and suggestions about this player.
```

Example prompt:

```
Player sees: <grass, water, cow>

Past action: <sleep>

Past sub-goals:
- eat cow
- collect stone
- place stone

Understanding score: <0.1>

Analyze the environment and the player's understanding capability,
then generate concise summaries and suggestions about this player.
```

Example output:

```
The player can't understand past sub-goals well. To help the agent
 learn the skill of eating cow more quickly, please provide more
detailed guidance.
```

### B.2  PROMPTS FOR LLM IN RLADAPTER

System message:

```
You are a professional game analyst. A player is playing a game
similar to Minecraft. Available actions are:
<move_left, move_right, move_up, move_down, do, sleep, place_stone
, place_table, place_furnace, place_plant, make_wood_pickaxe,
make_stone_pickaxe, make_iron_pickaxe, make_wood_sword,
make_stone_sword, make_iron_sword>.

You will get analysis about this player from another analyst, and
you will be asked to provide the next sub-goals for this player.
```

Example prompt:

```
Player sees: <grass, water, cow>

Past action: <sleep>

Past sub-goals:
- eat cow
- collect stone
- place stone

Analysis: <The player can't understand past sub-goals well. To
help the player learn the skill of eating cow more quickly, please
 provide more detailed guidance.>

Based on the provided information, suggest 3 sub-goals that the
player should accomplish next.
```

Example output:

```
- find cow
- move to cow
- eat cow
```

### B.3 PROMPTS FOR RLADAPTER W/O ADAPTER MODEL

System message:

```
You are a professional game analyst. A player is playing a game
similar to Minecraft. Available actions are:
<move_left, move_right, move_up, move_down, do, sleep, place_stone
, place_table, place_furnace, place_plant, make_wood_pickaxe,
make_stone_pickaxe, make_iron_pickaxe, make_wood_sword,
make_stone_sword, make_iron_sword>.

You will get necessary information, and you will be asked to
provide the next sub-goals for this player.
```

Example prompt:

```
Player sees: <grass, water, cow>

Past action: <sleep>

Past sub-goals:
- eat cow
- collect stone
- place stone

Based on the provided information, suggest 3 sub-goals that the
player should accomplish next.
```

Example output:

```
- collect stone
- make stone sword
- make stone pickaxe
```

### B.4    PROMPTS FOR STANDALONE GPT-4/GPT-3.5

`System message:`

```
You are a professional game analyst. A player is playing a game
similar to Minecraft. Available actions are:
<move_left, move_right, move_up, move_down, do, sleep, place_stone
, place_table, place_furnace, place_plant, make_wood_pickaxe,
make_stone_pickaxe, make_iron_pickaxe, make_wood_sword,
make_stone_sword, make_iron_sword>.

You will get necessary information, and you will be asked to
provide the next action for this player. You are only allowed to
choose the available action.
```

`Example prompt:`

```
Player sees: <grass, water, cow>

Past action: <sleep>

Based on the provided information, provide the next action for
this player.
```

`Example output:`

```
move_left
```

## C    CALCULATING ROGUE-L SIMILARITY

The ROGUE-L similarity is calculated as follows.

$$R_{lcs} = \frac{LCS(X,Y)}{m}, \tag{3}$$

$$P_{lcs} = \frac{LCS(X,Y)}{n}, \tag{4}$$

$$F_{lcs} = \frac{(1+\beta^2) R_{lcs} P_{lcs}}{R_{lcs} + \beta^2 P_{lcs}}, \tag{5}$$

where $LCS(X,Y)$ is the length of the longest common subsequence of $X$ and $Y$, $m$ is the length of $X$, $n$ is the length of $Y$, $\beta = \frac{R_{lcs}}{P_{lcs}}$, and $F_{lcs}$ is the ROGUE-L similarity. In our implementation, the threshold for ROGUE-L similarity is set to 0.7. Only when $F_{lcs} < 0.7$, it will be added to the fine-tuning data pool.

# D    CASE DETAILS

## D.1    AVOIDING COMBAT

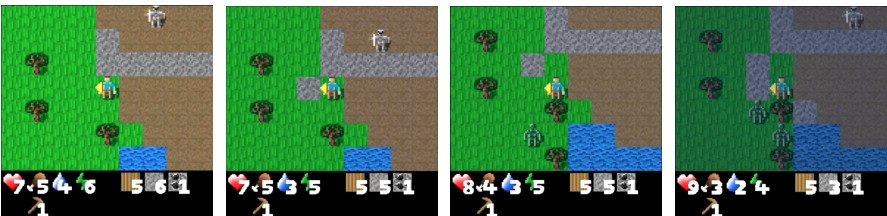

**Figure 4.** Case details of avoiding combat.

As shown in Figure 4, it is approaching night and the number of monsters is increasing. The agent starts early to strategically place stones in suitable terrain, successfully building a shelter that can keep the monsters outside and extend its survival time.

## D.2    RESOURCE PLANNING

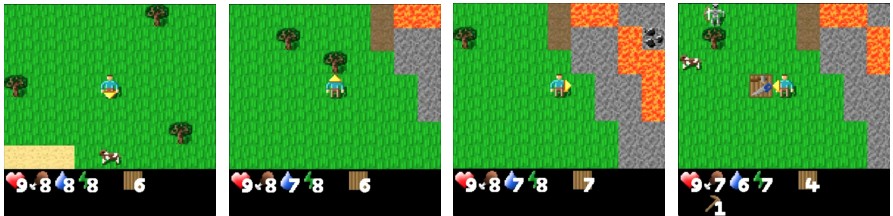

**Figure 5.** Case details of resource planning.

As shown in Figure 5, even though the agent has enough wood to make a workbench, its observations do not reveal abundant resources. Therefore, instead of rushing to make a workbench, it waits until more resources are discovered before making one nearby. This strategy can optimize the efficiency of resource collecting and item crafting.

# E    BROADER IMPACT

While the guidance generated by LLMs exhibits strong common-sense capabilities, there is a possibility that they might contain or produce harmful information. Though no such concerns were observed during evaluations in simulated environments like *Crafter*, it is imperative to address these potential risks when transferring RLAdapter to more open and real-world settings in the future. Mitigating these risks can be achieved by adding additional instructions in prompts, fine-tuning with curated data, and post-processing the generated text. Adopting these measures ensures that RLAdapter functions effectively and safely in its intended roles.

## F    COMPUTE RESOURCE DETAILS

The hardware resources we used are listed in Table 7. For each seed, the average GPU running speed is approximately 30K steps/hour.

**Table 7.** Computational resources for our experiments.

| CPU | GPU | RAM |
|-----|-----|-----|
| Intel Xeon 8280@2.7GHz | Nvidia A100 (40GB) | 256GB |
| Intel I9-12900K@3.2GHz | Nvidia RTX 3090 (24GB) | 128GB |

## G    LICENSES

In our code, we have used the following libraries which are covered by the corresponding licenses:

- Numpy (BSD-3-Clause license)
- Pandas (BSD-3-Clause license)
- PyTorch (BSD-3-Clause license)
- OpenAI Gym (MIT license)
- LLaMA 2 (LLaMA 2 license)

