# OpenReview forum: "RLAdapter: Bridging Large Language Models to Reinforcement Learning in Open Worlds"
_ICLR.cc/2024/Conference — ICLR 2024 Conference Withdrawn Submission_

### Official Review · Reviewer_mt27 · 2023-10-30

**Soundness:** 3 good
**Presentation:** 2 fair
**Contribution:** 3 good
**Rating:** 5
**Confidence:** 3

**Summary:**

The paper presents RLAdapter, a framework that enhances the collaboration between Large Language Models (LLMs) and Reinforcement Learning (RL) algorithms. It introduces an adapter model that acts as a bridge between the LLMs and RL agents, aiding in better guidance and improved learning in sparse-reward environments. The adapter model is fine-tuned with information generated during the RL agents' training process, making the LLMs more adaptable to downstream tasks without directly modifying the LLMs. The RLAdapter has been empirically evaluated in the Crafter environment, showing superior performance over state-of-the-art baselines and demonstrating common-sense behaviors not observed in baseline models​.

**Strengths:**

- Novelty and Clever Design: The introduction of an adapter model to bridge LLMs and RL algorithms is a novel idea. The adapter model helps in effectively summarizing and communicating the essential details from the RL agents to the LLMs.
- Empirical Evaluation: The empirical evaluation of RLAdapter in the Crafter environment and its comparison with several baselines demonstrate its effectiveness. The RLAdapter not only showed improved performance over state-of-the-art baselines but also exhibited agent behaviors grounded in common sense.

**Weaknesses:**

- Understanding Score: The paper uses an understanding score calculated based on the semantic similarity between a trajectory and the LLM’s sub-goals. However, it seems that this aspect is understudied in the paper. A more in-depth exploration and justification of the understanding score, possibly discussing its limitations, would have strengthened the paper. It’s unclear whether it indeed does what the authors intended.
- It’s unclear whether SFT is needed or even helpful. Perhaps few-shot-learning would be enough.
- In the ablation study, when removing the RLAdapter model, I believe the authors should have used the same prompts with the LLM. I was left thinking that perhaps the improved performance comes from a form of self-critique of the goal proposer before writing the next goal. The same could be achieved with prompting GPT.
- SPRING is mentioned quite few times, so I'd expect it to be better mentioned in "related work". In particular, examples of tight coupling with the environment.

**Questions:**

- Analysis of Adapter Model's Capability: Does the adapter model have the capability of generating more general goals apart from making the LLM give more detailed goals? An exploration of this aspect would provide a more comprehensive view of the adapter model's functionalities.
  * If the adapter model only makes the LLM give more detailed sub-goals, we could simply prompt the LLM to analyze whether the last sub goal seems to be achieved and give more detailed sub-goals if not.
- Similar to the point above, it seems like the adapter model could be replaced by the LLM itself? Just querying it twice, one for analysis and another for sub-goal.
- Detailed Analysis: A more detailed analysis, such as an ablation study, on the understanding score, and how it influences the model's performance, could be beneficial.
- [Minor] The use of training steps in Table 1 as a metric might not be the most relevant. Consideration of other metrics like FLOPs could provide a more standardized comparison across different models. I understand this is probably not feasible given that OpenAI inference FLOPs are not public.

---

### Official Review · Reviewer_aWpB · 2023-10-30

**Soundness:** 2 fair
**Presentation:** 3 good
**Contribution:** 2 fair
**Rating:** 3
**Confidence:** 4

**Summary:**

This paper focuses on solving RL tasks with the presence of an LLM that receives environmental descriptions and generates goals. This paper proposes RLAdaptor, which is a language model that takes as input a description of the current state (including other relevant information) and a scalar score that indicates how well the policy follows the intended goal (by computing a similarity score between the goal and a manually summarized sequence of gameplay). The RLAdaptor then generates texts that are given to the LLM as additional information.

The authors claim that RLAdaptor can effectively "steer" the LLM to provide more useful information to the RL algorithm. In turn, the RL policy collects data that can be used by the RLAdaptor.

Experiments in the Crafter environment demonstrate better performance compared to the baselines.

**Strengths:**

The idea of adding a lightweight model to adjust the behavior of LLMs to provide better guidance is very interesting since directly fine-tuning an LLM is costly. The proposed RLAdaptor model uses a score that indicates whether the policy successfully executes the intended subgoal.

The paper is clear in general and the proposed method achieves some improvements over the baselines.

**Weaknesses:**

Despite the name RLAdaptor, it seems that the proposed method is only tested in the text-Crafter environment, where the action space is a set of high-level movements (e.g., collect certain resources). In this case, the goal-conditioned policy $\pi (a | s, g)$ could be very trivial as it only needs to "translate" g into a valid text-based action a (or identically, an index to the corresponding text). Therefore, although the paper claims that RLAdaptor can guide LLM to be more "helpful" to the RL algorithm, the results in the paper are insufficient to justify this claim. It would be nice to show the effectiveness of the adaptor module in environments that require more low-level control.

Next, it seems that RLAdaptor has a similar performance compared to SPRING, an LLM-based agent that merely uses zero-shot prompting and no training. Specifically, in Table 1, given the high variance, RLAdaptor is not significantly better than SPRING. Given that RLAdaptor requires multiple rounds of training and data-collection, its advantage compared to zero-/few-shot prompting methods is diminished.

In addition to the environment state descriptions, the only information acquired by the adaptor model is the understanding score. It would be natural to compare the proposed training strategy with the preference learning pipeline where the understanding score is considered as a type of preference.

**Questions:**

The text-crafter environment seems to be insufficient to justify the effectiveness of RLAdaptor (see weakness).

The results are only compatible / slightly better than a zero-shot LLM prompting method.

How does the proposed training technique connect to the preference-based learning paradigm (e.g., RL from human feedback)?

---

### Official Review · Reviewer_1niL · 2023-10-31

**Soundness:** 2 fair
**Presentation:** 4 excellent
**Contribution:** 3 good
**Rating:** 3
**Confidence:** 2

**Summary:**

The paper proposes an algorithm for finetuning language models to assist agent learning. Specifically, they finetune a small LLM (llama2 7b) to provide part of the prompt for a bigger, frozen LLM (GPT3.5 or 4). Specifically, they follow this approach:
1. The adapter model is fed the current state (described in language), past subgoals, and a score estimating how well the agent was able to understand and complete the past list of subgoals.
2. The adapter model summarizes the agent's understanding and provides suggestions.
3. The adapter model's output is fed into the frozen LLM, along with the state description and past subgoals. The frozen LLM predicts the next subgoals.
4. The subgoals are embedded and fed to the agent


They test this in the Crafter environment and show slightly improved results over SPRING, another algorithm which requires expert knowledge of the environment, and significant improvements over all other baselines. Their ablations show that including the adapter model improves performance.

=========
Note on my overall score: I chose low-confidence b/c I didn't understand some core parts of the paper, but I did read the details and appendices thoroughly.

**Strengths:**

- This is an important and open problem. It's well-known that frozen LLMs can be used as agents/to assist agent learning, but it's also well-known that these frozen language models often fail to adapt to specifics of the environment and are unable to learn from experience. This is an important problem setting, and this is a unique approach!
- The paper does extensive comparisons to other methods, both standard RL methods and LLM-based methods.
- The paper ablates its key component - the finetuned adapter model - and finds that the adapter provides substantial performance improvements.
- The RLAdapter algo's performance significantly improves over all baselines except SPRING. I think this is fine since SPRING makes stronger assumptions - namely, being able to extract a graph of tasks from the paper that proposed the environment.

**Weaknesses:**

- The paper tests only on one environment, Crafter, making it unclear how easily this method applies to other environments. I think this is not a big deal since Crafter is a challenging environment, but I wish the paper explicitly spelled out the requirements an environment/task must have for the algorithm to be applicable (e.g. must be able to provide language observations)
- The main weaknesses are some confusions about the method, which I describe in the section below. I did not understand the core algorithm. My score could improve significantly if this was clarified.

**Questions:**

**Confusions from section 3.4 and Algorithm 1**
- What trajectory is being passed into the understanding-scoring function? In algorithm 1, the trajectory is being sampled from $B_{t+1}$, a buffer which contains all past trajectories. But the trajectory is being compared with $g_t$ the current goal. I'd expect the current goal to be compared to the current partial-trajectory (which is I think what section 3.4 suggests, though that paragraph is confusing).
- The "prompt" function seems to refer to two separate functions - one which constructs a prompt for $M_{ada}$ and one which constructs a prompt for $M_{LLM}$. I'd suggest making these "prompt_a" and "prompt_l".
- The line "Then we construct a pair of linguistic data l = [prompt(B, u_new), prompt(B, c)]" is confusing. Should the second item be $c$ since that's the output of the adapter model? I think this is what Algorithm 1 shows.
- Section 3.4 should use subscripts, like in the algorithm box. Also, switch all the "new" subscripts to time indices for clarity. (Currently, $g_{new}$ and $\tau_{new}$ appear without being defined first.)
- I don't understand the intuition behind the (input, output) pairs used for finetuning the adapter model. Here's an example that confuses me: Let's say that the agent completely misunderstood the current instructions, resulting in a low $u_t$. Then, the adapter model produced an analysis $c_t$ that said something like "simplify the instructions". Let's say that with these instructions, the LLM produced a good $g_t$ and the agent succeeds, resulting in a high $u_{t+1}$. Then, the training point added to $D$ would be "(prompt($B_t$, $u_{t+1})$, $c_t$)", which would mean the prompt indicates that the agent understood the instructions (high $u_{t+1}$), but the desired output is "simplify the instructions", which doesn't make sense in context. Am I misunderstanding this?
- The paper says "Once updated, new trajectories yield a refreshed understanding score." What update is this referring to, and which new trajectories? $u_{t+1}$ is computed every timestep. Is the only difference between the trajectories used to compute $u_t$ and $u_{t+1}$ one additional timestep?
- Section 3.4 says the adapter is improved with "supervised fine-tuning", but the referenced paper includes an extra contrastive loss. Are you using this loss? If so, please highlight that this is *not* standard SFT and explain why this approach was selected.
- It's not clear why filtering by the ROGUE-L score produces trajectories which are good for updating the adapter model using SFT. The ROGUE-L score seems like it would introduce diversity, but I don't understand what part of this algorithm is filtering for trajectories which contained high-quality $c_t$ analyses.

**Confusions about the benefits of this algorithm**
- Table 3 shows "common-sense behaviors" such as placing stones in front of zombies. Were these behaviors proposed by the LLM, by the adapter model, or not at all? I would be surprised if either of the language models could specify this, since they are given only a high-level description of the scene saying what objects are visible and presumably didn't know where the agent was positioned relative to zombies and blocks. If the language models didn't propose these goals, then it seems like a stretch to claim the RLAdapter incentivizes these behaviors -- wouldn't a normal RL baseline also learn these since they lead to higher reward?
- I wonder how much of RLAdapter's benefit comes from the RL finetuning vs how much comes from having an understanding score. I'd like to see a baseline that includes the understanding score into the prompt for $M_{LLM}$.
- I don't see anything in the prompts for $M_{ADA}$ or $M_{LLM}$ which incentivizes the agent to pursue a diverse set of goals. Do you run into issues where the LLM proposes the same goals over and over again (e.g. things like "chop grass" since grass is visible in every frame)?
- Does the agent actually follow the goals provided? (I ask this because it seems like the agent's score is unaffected by how well it follows goals, and the constantly-changing adapter model seems like it would result in the subgoals given in particular situations changing often, which might make it harder for an agent to learn what the language means.)
- Some analysis of (a) how often the provided goals are accomplished, (b) how often the understanding module correctly indicates the agent's proficiency with a goal, and (c)how often the adapter model gives a useful suggestion would be nice.

**Miscellaneous other suggestions**
- Somewhere in the appendix, include the specifics of how the language observations were formatted, as well as how trajectories are represented (for the understanding model $u$).
- The ELLM comparison is not quite apples-to-apples since ELLM made a few env modifications (see section 4.1 in their paper). This is fine since I suspect ELLM would perform worse without the changes (and also the performance gap between ELLM and RLAdapter is pretty big), just maybe mention the changes in your appendix somewhere.
- I'd suggest having a paragraph describing SPRING since it is your main baseline and since the paper mentions details of the algo (e.g. that it relies on expert knowledge).

---

### Official Review · Reviewer_6tyY · 2023-10-31

**Soundness:** 2 fair
**Presentation:** 3 good
**Contribution:** 2 fair
**Rating:** 3
**Confidence:** 4

**Summary:**

The paper argues although LLMs contain a fair amount of high level knowledge and could potentially significantly improve the performance of RL policies. At the same time, LLMs are not adapted to the current environment and this lack of understanding can hinder their ability to help RL agents. The paper argues that fine-tuning the whole LLM is costly, and therefore propose to use an adapter that would provide additional information to the LLM. This adapter is essentially a second LLM that is being trained to predict the RL agent's capabilities. The authors perform experiments on the open world Crafter environment.

**Strengths:**

* The idea to avoid complete fine-tuning of the LLM is very reasonable and is the backbone of popular methods such as LoRA.
* The idea to have a single scalar evaluate the RL agent's understanding of the LLMs guidance (or in general its competence in the game) is a promising avenue
* The authors provide qualitative and quantitative results.

Fine-tuning a large model is costly and likely will destroy some of its prior knowledge. We need to find ways to bridge the gap between the model's understanding and the reality of the RL agent. This paper propose one way to do so.

**Weaknesses:**

* The claims of performance made in the abstract, introduction and experiments section are misleading. The paper claims that the methods performs better than the baselines (and even highlight their score in Table 1), yet this advantage is not statistically significant since the standard deviations heavily intersect. In Figure 3 there is not even an inclusion of error bars.
* The method builds on the idea of providing an analysis of the agent's capacities to the LLM which selects subgoals. This has been investigated a few ways in recent papers. A baseline that includes this kind of feedback (without the RLAdatper) would be important).
* There no analysis whether the proposed cosine similarity really captures the RL agent's understanding of the task

With the advent of LLMs there has been numerous recent paper that lack scientific rigour, for example by excluding error bars or hiding a lot of heavy engineering in the appendix. This is very detrimental to the community as we have worked a lot of making RL statistically significant [1, 2]. The paper currently makes claim ("RLAdapter surpasses the SOTA baselines") that simply are not statistically significant and the paper additionally does not provide errors in some plots. This is a serious concern for science. Moreover, the paper refers to the SPRING baseline as requiring "expert-level data dependency" - this is quite hand wavy. It might imply that expert-level data is expert level trajectories, which is clearly not the case.

At the core of the method is the idea that an LLM will choose better subgoals if it given more context about the RL agent's abilities and understanding. The specific instantiation of this idea is through an understanding score. First, there is no experiment investigating whether the understanding score really captures the RL agent's capabilities. Second, the general idea of giving the context of which task an RL agent can achieve has been explored in various in the literature, for example  [3]. An important baseline would be to simply provide context about which subgoals were achieved by the RL agent. This would essentially evaluate the main contribution of the paper: the use of an understanding score to guide the selection of subgoals.

**Questions:**

How is the training proceed of the Adapter actually done? There is no equation about this in the paper. Moreover in section 3.4 refers to a pair of linguistic data where the prompt function takes different arguments, sometimes the second argument is "u_new" and sometimes it's "c". Why the use of ROGUE-L similarity?

"enables the adapter model to refine its self-awareness" Is the Adapter really self-aware?

The abstract claims that the adapter is lightweight, yet in the main paper this is revealed to be GPT3.5. How is this model a lightweight model? This does not sound like the right way to describe it.

"RLAdapter demonstrates behaviors that align with human common sense" This is very counterintuitive to me. To play the game of Crafter you have to fight zombies at least once in a game, which would render the success rate of this task to 100%.


"These behaviors are not observed or reported in other baselines" This is unfair to the baseline as it correlate not reporting with not observing something. If the code is open sourced for some of the baselines a good thing to do would be to verify.

=========================================================================

[1] Deep Reinforcement Learning that Matters. Henderson et al., 2017

[2] Deep Reinforcement Learning at the Edge of the Statistical Precipice, Agarwal et al., 2021

[3] OMNI: Open-endedness via Models of human Notions of Interestingness, Zhang et al., 2023